# Sustained In-Vivo Release of Triptorelin Acetate from a Biodegradable Silica Depot: Comparison to Pamorelin^®^ LA

**DOI:** 10.3390/nano11061578

**Published:** 2021-06-16

**Authors:** Ari-Pekka Forsback, Panu Noppari, Jesse Viljanen, Jari Mikkola, Mika Jokinen, Lasse Leino, Simon Bjerregaard, Camilla Borglin, Janet Halliday

**Affiliations:** 1DelSiTech Ltd., 20520 Turku, Finland; aripekka.forsback@delsitech.com (A.-P.F.); jesse.viljanen@delsitech.com (J.V.); jari.mikkola@delsitech.com (J.M.); lasse.leino@delsitech.com (L.L.); 2Department of Chemical and Metallurgical Engineering, Aalto University, 02150 Espoo, Finland; 3Ferring Pharmaceuticals, 2770 Kastrup, Copenhagen, Denmark; simon.bjerregaard@ferring.com (S.B.); camilla.borglin@ferring.com (C.B.); 4Ferring Controlled Therapeutics, East Kilbride G74 5PB, Scotland, UK; janet.halliday@ferring.com

**Keywords:** controlled release, silica hydrogel, silica microparticles, silica nanoparticles, triptorelin

## Abstract

Triptorelin acetate was encapsulated into silica microparticles by spray-drying a mixture of colloidal silica sol and triptorelin acetate solution. The resulting microparticles were then combined with another silica sol containing silica nanoparticles, which together formed an injectable silica-triptorelin acetate depot. The particle size and surface morphology of the silica-triptorelin acetate microparticles were characterized together with the in vitro release of triptorelin, injectability and rheology of the final injectable silica-triptorelin acetate depot. In vivo pharmacokinetics and pharmacodynamics of the silica-triptorelin acetate depot and Pamorelin^®^ were evaluated and compared in Sprague-Dawley male rats after subcutaneous administration. Serum samples up to 91 days were collected and the plasma concentrations of triptorelin and testosterone were analyzed with ultraperformance liquid chromatography-tandem mass spectrometry (UPLC-MS/MS). In vivo pharmacokinetics showed that injections of the silica-triptorelin acetate depot gave 5-fold lower Cmax values than the corresponding Pamorelin^®^ injections. The depot also showed a comparable sustained triptorelin release and equivalent pharmacodynamic effect as the Pamorelin^®^ injections. Detectable triptorelin plasma concentrations were seen with the depot after the 91-day study period and testosterone plasma concentrations remained below the human castration limit for the same period.

## 1. Introduction

Silica is a well-known material in the field of controlled drug delivery. It has been used as a matrix material to delivery different types of active pharmaceutical ingredients (API) and silica has been manufactured into various dosage forms (e.g., microparticles, fibers and monolithic implants) [1]. Depending on the type of silica used to deliver an API, the incorporation of the API as well as the release mechanism of the API can vary. For example, with mesoporous silica the incorporation of the API and its release are mainly based on the pore structure and/or surface modification of the pore walls [2,3]. There are some general benefits with mesoporous silica, such as high loading capacity and possibility to control and modify the pores with micro- and nanoscale particles. On the other hand, pore-dependent release often results in diffusion-controlled API release mechanism. Furthermore, incorporating different types of APIs can prove to be challenging due their size e.g., peptides and proteins or other larger therapeutic agents, such as viral vectors.

The other common approach for preparing silica for controlled drug delivery is the direct encapsulation/embedment of API during sol-gel processing, which involves incorporating an API into a colloidal silica nanoparticle solution (i.e., silica sol). In practice, with the silica sol-gel processing method, there is no upper limit for the size of API that can be encapsulated/embedded into the nanoscale silica matrix. When the encapsulation/embedment into the silica matrix is effective, the API release is mainly based on matrix erosion (i.e., silica dissolution). This controlled release driven by matrix erosion can be achieved when the resulting pore structure is smaller than the API or the pore size is so small that the diffusion rate of the API is neglectable in comparison to the dissolution rate of the matrix. The dissolution rate and subsequent release of the API is mainly controlled by adjusting and controlling the chemical structure of the silica matrix (i.e., controlling the number of free OH-groups and degree of condensation) [4,5,6]. The challenge, in the direct encapsulation/embedment method, is to match the processing conditions of silica with the conditions that are suitable for the API in question. Naturally, these processing conditions are commonly more challenging for biological drugs compared to small-molecule drugs. In addition, there are many different types of biological drugs, which are different in size, structure, chemical and electrochemical properties. Therefore, the development of a platform-like technology with readily adjustable properties is preferable. 

Development of minimally invasive drug delivery devices has also been one of the general aims in the field of drug delivery for quite some time. Recently the development has been accelerated by the new biological drugs for intraocular release [7], which has given further incentive to develop injectable dosage forms for other routes of drug administration (e.g., intramuscular, and subcutaneous). In terms of needle size and pain experienced by the patient, minimally invasive injections are still a subjective concept, but in general 22–25 G needles (outer diameter 0.7176–0.5144 mm) are adequate when injecting volumes equal to or less than 1 mL relatively infrequently (e.g., administering vaccinations subcutaneously). Logically patients experience less pain as the needle diameter decreases [8], and some administrations routes, such as intravitreal delivery, require significantly smaller needle diameters to increase patient compliance for injections [9].

Injectable dosage forms are often based on different types of microparticles in suspensions [10,11], or on hydrogels [12]. Several microparticle-based formulations have been developed for Triptorelin and its pharmaceutically suitable salts. These formulations are clinically available and provide a 1-, 3- or 6-month sustained release of Triptorelin for prostate cancer treatment [13]. One of them is Pamorelin^®^, a PLGA-based formulation containing triptorelin pamoate, administered with a 20 G needle (outer diameter 0.908 mm). Injectable suspensions and hydrogels are often based on organic polymers, but there are also inorganic formulations available (e.g., silica-based dosage forms). 

The direct encapsulation/embedment of API in silica has been used for several different dosage forms, but a minimally invasive injectable depot formulation has been developed [14]. Said silica-based depot formulation comprises of silica microparticles embedded in a silica hydrogel containing nanoscale silica particles. The resulting depot is an injectable particle-reinforced composite, where the silica microparticles are joined together by silica nanoparticles to form a stable three-dimensional gel structure that flows when sheared. It has also been shown that properties of this type of silica-based composite can be adjusted to result in zero-order release [15]. The silica-based composite is non-flowing and stable at rest (preventing sedimentation of microparticles), but injectable as shear force is applied (e.g., during injection). These viscoelastic properties of the composite make it possible to have the dosage form stored in a prefilled syringe in a ready-to-use format. Not requiring any preparations prior to injection.

The aim of the study was to adapt the silica-based composite, comprising of silica microparticles embedded in a silica hydrogel, for a controlled and sustained delivery of triptorelin acetate (a decapeptide). Triptorelin acetate was chosen to show the potential of the silica technology for controlled delivery of peptides due to good pharmacodynamic and pharmacokinetic modeling. Hereafter, referred to as silica-triptorelin acetate depot. This included optimizing the injectability of the silica-triptorelin acetate depot, so that it could be administered using 22–25 G needles ensuring a minimally invasive delivery of the API. The silica-based depot was characterized both in vitro and in vivo. The in vitro studies include the characterization of the microparticles embedded into the hydrogel (particle size distribution and SEM-imaging). The rheological properties and injectability of the final silica-based depot were characterized. The controlled and sustained delivery of triptorelin acetate was analyzed both in vitro and in vivo. In vitro release was characterized by conducting dissolution experiments at in sink conditions for silica dissolution and triptorelin acetate. In the vivo study, commercially available Pamorelin^®^ was evaluated alongside the silica-based depot after subcutaneous injections in Sprague-Dawley male rats. The in vivo release of both formulations was investigated by conducting a pharmacokinetic study including the analysis of plasma testosterone levels to determine the pharmacodynamic effect of the released triptorelin.

## 2. Materials and Methods

Reagent grade tetraethyl orthosilicate (TEOS) was purchased from Sigma-Aldrich (St. Louis, MO, USA) and analytical grade solutions of 0.1 M hydrochloric acid (0.1 M HCl, Merck Titripur^®^) and 0.1 M sodium hydroxide (0.1 M NaOH, Merck Titripur^®^) were purchased from VWR Chemicals (Radnor, PA, USA). Triptorelin acetate was supplied by Ferring Pharmaceuticals and Pamorelin^®^ 11.25 mg, batch N19449 (Ipsen Pharmaceuticals, Paris, France), was purchased. Testosterone standard for bioanalysis was from TCI (Tokyo, Japan) and internal standards alarelin acetate (Ark Pharma, Arlington Heights, IL, USA) and deuterated testosterone (testosterone-D3, Cerilliant, Round Rock, TX, USA) were purchased from Sigma-Aldrich. All other materials and reagents used were of analytical grade and purchased from Sigma-Aldrich/VWR unless otherwise specified. 

The silica microparticles were manufactured by spray-drying a colloidal silica solution (silica sol) mixed with an alkaline triptorelin acetate aqueous solution. The alkaline triptorelin acetate solution was prepared by dissolving triptorelin acetate in deionized water and adjusting the pH of the solution to 10.1 with a 0.1 M NaOH solution. The final triptorelin acetate concentration in the solution was 0.7 mg/mL. The second component, the silica sol, was produced by hydrolysis of TEOS in deionized water using 0.1 M HCl as a catalyst. The molar water:TEOS:HCl ratio in the sol was 4:1:0.03. The hydrolysis reaction was carried out under strong mixing at room temperature and after the hydrolysis reaction the resulting silica sol was cooled down in an ice water bath. 

The spray-drying process was performed as a continuous feed process where the triptorelin acetate solution and the cold silica sol was fed into a tube reactor using a peristaltic pump and silicon tubing. The tube reactor was connected to the spray-dryer and the pumping rates of the two solutions were adjusted in way that the pH of the combined solution in the reaction tube was 4.6 and the triptorelin acetate solution feed rate was ten-fold compared to the silica sol feed rate. The spray-dryer was a Buchi B-290 (Büchi AG, Flawil, Switzerland) and the aspirator air flow rate was 100%, atomization air flow was 670 L/h, total feed rate was 5.6 mL/min, and the inlet and outlet temperatures were 120 °C and 73 °C, respectively. 

To produce the injectable silica-triptorelin acetate depot, the silica-triptorelin acetate microparticles were incorporated in a R400 silica sol (i.e., molar ratio of water to TEOS was 400) and the resulting microparticle-sol suspension was stabilized into a semi-solid gel at room temperature for three days. The R400 silica sol was produced by hydrolysis of TEOS in deionized water, using a 0.1 M HCl solution as a catalyst, under strong mixing at room temperature. After the hydrolysis reaction, the pH of the R400 silica sol was set to 5.9 with 0.1 M NaOH solution. Then the silica-triptorelin acetate microparticles were suspended in the R400 silica sol with a 0.75:1 ratio (*w*/*v*). The homogenous microparticle-sol suspension was then filled into disposable 1-mL BD Luer-Lok syringes, Catalogue no. 309628, (Becton, Dickinson and Company, Franklin Lakes, NJ, USA) and stabilized into semi-solid gel within the syringes at room temperature for three days using a custom-made tube rotator (DelSiTech, Turku, Finland). Noteworthy is that without the nanoparticles (derived from the R400 silica sol) the aqueous microparticle suspension is unstable at rest resulting in poor injectability. Just by incorporating ca. 0.5 wt.% silica nanoparticles is enough to transform the unstable microparticle suspension into a stable semi-solid injectable gel. 

The total silica content of the silica-triptorelin acetate microparticles and the silica-triptorelin acetate depot was measured by completely dissolving 10–30 mg samples in 50 mL of 0.5 M NaOH solution for three days at 37 °C. The silica concentrations of the samples were measured with a UV/VIS spectrophotometer (JASCO Corporation, Tokyo, Japan) by analyzing the molybdenum blue complex absorbance at λ = 820 nm [16]. 

The total triptorelin content was measured indirectly with a peptide hydrolysis method by dissolving silica-triptorelin acetate microparticle and silica-triptorelin acetate depot samples in strong alkali, simultaneously hydrolyzing triptorelin into amino acids. The hydrolysis was done in 3 M NaOH solution containing 5% 2,2′-thiodiethanol and 5 mg/mL amino acid mixture (arginine, histidine, glutamine, glycine, lysine and methionine) at 100 °C for 24 h after which the tryptophan was quantified with a 1260 Infinity HPLC (Agilent Technologies, Santa Clara, CA, USA) connected to a multiple wavelength detector (at λ = 280 nm). The chromatographic separation was obtained on a Waters Symmetry C18 3.5 µm, 4.6 × 150 mm HPLC column. The mobile phase A was water and formic acid in a ratio of 1000:1 (*v*/*v*) and the mobile phase B was acetonitrile and formic acid in a ratio of 1000:1 (*v*/*v*). The injection volume was 10 µL, the column temperature was 30 °C and the flow rate was 1.0 mL/min. Mobile phase B (%) was kept constant at 8% for 5.5 min and then gradually increased to 90% over 1.5 min. After 1-min washout period, mobile phase B (%) was reduced back to 8%. Control and calibration samples were prepared with the triptorelin dissolution test buffer. In vitro degradation of silica and resulting release of triptorelin from the silica-triptorelin acetate depot was measured in 50 mM TRIS buffer pH 7.4 containing 0.01 % (*v*/*v*) TWEEN^®^ 80 at 37 °C. The silica concentration in the dissolution buffer was kept at in sink conditions (free dissolution of the silica matrix) by dissolving 10–30 mg samples in 50 mL of the dissolution buffer. The dissolution buffer was changed to fresh medium at every sampling time point to keep silica concentrations below 30 ppm (at in sink condition). The dissolution studies were conducted for 7 days in a shaking water bath (60 strokes/min) at 37 °C (Julabo Gmbh, Seelbach, Germany). The dissolved silica was measured as described earlier. The released triptorelin was quantified by HPLC (Agilent Technologies 1260 Infinity) with a Waters XBridge C18 2.5 µm, 3.0 × 20 mm column. Herein, the mobile A phase consisted of water and trifluoroacetic acid in a ratio of 1000:2.5 (*v*/*v*) and the mobile phase B consisted of acetonitrile and trifluoroacetic acid in a ratio of 1000:2.5 (*v*/*v*). The absorbance was detected at 220 nm, the injection volume was 40 µL, the flow rate was 1.0 mL/min and the column temperature was 80 °C. Mobile phase B (%) was first gradually increased from 15% to 50% over 1.5 min followed by another increase from 50% to 99% over 1 min. After 0.5 min washout period, mobile phase (%) was reduced to 15%.

The animal study was conducted by qualified animal technicians in the Central Animal Laboratory of Turku University (Finland) in compliance with the guidelines of the National Laboratory Animal Board of Finland. Specific Pathogen Free (SPF) Sprague-Dawley male rats (RjHan:SD, CD^®^ Rats) 9 weeks old and 250–300 g in weight, were subcutaneously administered 5 mg/kg or 10 mg/kg of triptorelin acetate. Triptorelin and testosterone plasma concentrations of four groups (I, II, III and IV) of 6 animals (n = 6) after administration of Pamorelin^®^ and the silica-triptorelin acetate depot were compared. 

On the day of administration, Pamorelin^®^ (powder and solution for injection) was prepared according to the package leaflet instructions: one vial of Pamorelin powder (poly (d,l-lactide-co-glycolide), mannitol, carmellose, sodium, polysorbate 80) was reconstituted in 2 mL of solvent (water for injection) provided in two ampules. The solvent was added to the powder using a syringe and injection needle provided in the package. Once the powder was reconstituted the resulting suspension was drawn into the syringe and the needle was changed to a sterile injection needle, also provided in the packaging, before administration. Pamorelin^®^ was administered using a 20 G × 1” needle (Terumo^®^, Tokyo, Japan), whereas the silica-triptorelin acetate depot was administered using a 23 G × 1” needle (Terumo^®^, Tokyo, Japan). 

Group I received 1.13 mg of triptorelin acetate in 200 µL injection of Pamorelin^®^. Group II received 1.27 mg of triptorelin acetate in 100 µL injection of the silica-triptorelin acetate depot. Group III received 2.54 mg of triptorelin acetate in 200 µL injection of the silica-triptorelin acetate depot. Group IV received 2.26 mg of triptorelin acetate in 400 µL injection of Pamorelin^®^. Before and after dosing, blood samples were collected for 91 days and the plasma, from the blood samples, was frozen and stored at −20 °C for later analysis. 

The triptorelin and testosterone concentrations of the plasma samples were simultaneously analyzed with UPLC-MS/MS (Waters Corporation, Milford, MA, USA). Sample preparation included solid-phase extraction (SPE) using HLB SPE plate (µElution plate 30 µM, Waters). 40 µL of rat plasma sample was mixed with 400 µL of ultrapure water and 40 µL internal standard solution containing 20 ng/mL of alarelin acetate and testosterone-D3 (16,16,17) and mixed. The SPE plate was first equilibrated by passing first 1 mL of methanol, and then 1 mL of water through each well. The samples were transferred onto the HLB plate and aspirated in the plate by applying 15 inHg vacuum for 10 min. The wells were then washed with 1 mL (60:40 Methanol-water) solution, followed by elution of the compounds with 1 mL of methanol containing 0.01% formic acid. The samples were transferred onto the HLB plate aspirated in the plate by applying 15 inHg vacuum for 10 min and evaporated to dryness in nitrogen flow and reconstituted into (60:40) methanol-water containing 0.01% formic acid. 

The chromatographic separation was obtained on a Waters Acquity BEH C18 (50 × 2.1 mm, particle size 1.7 µm) column using Acquity UPLC connected to Xevo TQ-S triple quadrupole mass spectrometer (Milford, MA, USA) from Waters. The mobile phase A was water and formic acid in a ratio of 1000:1 (*v*/*v*) and the mobile phase B was acetonitrile and formic acid in a ratio of 1000:1 (*v*/*v*). The injection volume was 4 µL, the column temperature was 40 °C and the flow rate was 0.5 mL/min. Mobile phase B (%) was kept constant at 2% for 0.5 min and then gradually increased to 95% over 2.0 min. After 0.5-min washout period, mobile phase B (%) was reduced back to 2%. Capillary voltage used in the measurement was 1.0 kV (ESI+). The internal standards for testosterone and triptorelin were testosterone-D3 and alarelin, respectively. Quantitation of triptorelin was performed by acquiring MRM 656 (IS:584) > 249 (CE 26) in positive polarity with a cone voltage of 30 V. Testosterone was quantitated by acquiring MRM 289 (IS: 292) > 97.2 (CE 22). Cone voltage was 30 V. 

The standard samples were prepared into female rat plasma by spiking the matrix into concentrations 0.1–5000 ng/mL of the analytes and otherwise treated as the samples. The quality control samples were prepared and treated identically with concentrations of 3, 30, 300, and 3000 ng/mL. To increase qualified sensitivity from previous prequalification, six Lower limit of quantitation (LLOQ) level quality control (QC) samples at 0.2 ng/mL for testosterone, and at 0.5 ng/mL for triptorelin, were added into the first samples run to prove the increased sensitivity of the method. Calibration curves were generated by quadratic fitting using weighting 1/x, resulting in R^2^ values all above 0.991 for both testosterone and triptorelin in both analytical runs, with respective calibrator ranges of 0.5–5000 ng/mL for triptorelin and 0.2–5000 ng/mL for testosterone. The lower limit of detection of the triptorelin and testosterone bioanalytical methods were 0.2 and 0.1 ng/mL, respectively. Pharmacokinetic parameters based on the bioanalytical data were calculated by Phoenix WinNonlin software (Certara Companies, Princeton, NJ, USA) using models Plasma Data, Extravascular Administration and Sparse Sampling. Calculation method used was Linear Trapezoidal with Linear Interpolation. 

After necropsy on day 91, the remnants of the silica-triptorelin acetate depots and Pamorelin^®^ were excised from the rats, frozen and stored at −20 °C before being analyzed for their triptorelin content. The silica-triptorelin acetate depot remnants were dissolved in 50 mL of 50 mM glycine buffer solution (pH 9.4), whereas the Pamorelin^®^ remnants were immersed in 50 mL of ethanol. The dissolutions were carried out at 37 °C in a shaking water bath (60 strokes/min). The glycine buffer was periodically refreshed during sampling to maintain the free dissolution of the silica matrix (in sink conditions). The triptorelin concentration of the samples was measured with the HPLC method described earlier. The dissolution test was conducted until the release of triptorelin was not detectable by the HPLC method described previously. 

Particle size distribution (PSD) and surface morphology of the silica-triptorelin acetate microparticles were characterized by laser diffraction and scanning electron microscopy (SEM), respectively. For the PSD measurements a HELOS 2370 (Sympatec, Calusthal-Zellerfeld, Germany) instrument was used and for the SEM imaging, a LEO Gemini 1530 SEM (Zeiss, Oberkochen, Germany) with a Thermo Scientific UltraDry silicon drift detector (SDD)(Waltham, MA, USA) was used. 

Rheological properties, such as viscoelasticity and thixotropy, of the silica-triptorelin acetate depot were characterized with a Haake RheoStress 300 rheometer (ThermoFisher Scientific, Waltham, MA, USA) using a parallel plate (d = 20 mm) measuring geometry. The dynamic viscosity and thixotropic behavior were measured by linearly increasing the shear rate from 1 s^−1^ to 1500 s^−1^ within 30 s, then decreasing the shear rate back to 1 s^−1^ within 60 s. The measuring gap was 0.3 mm. The viscoelastic properties (G’, G″ and tan δ) were studied with oscillatory measurements within the linear viscoelastic range of the samples with a measuring gap of 1.0 mm. The samples were studied under controlled deformation (γ < 0.002) within a frequency range of 0.1–10 Hz. All rheological measurements were performed at 25 °C.

The injectability was estimated manually using a 25 G × 1” hypodermic needle. The needle was attached to the prefilled syringe and then the syringe was emptied. The injected volume was 400 µL and the injectability was evaluated as pass or fail. The sample passed if the prefilled syringe could be emptied with a single steady continuous push of the plunger.

The endotoxin content of the silica-triptorelin acetate depot was analyzed in accordance with EP 2.6.14. Method B (gel-clot method; semiquantitative test), using Pyrogen^®^ 06 Plus kit (Lonza Inc., Morristown, NJ, USA). The measured endotoxin units (EU) were <0.06 EU/mg, which is below the Ph.Eur. limit of <0.3 EU/mg.

## 3. Results & Discussion

### 3.1. Characterization of Silica-Triptorelin Microparticles

The triptorelin payload in the silica-triptorelin microparticles was 3% (*w*/*w*). Particle size distribution and SEM images of the silica-triptorelin microparticles, used in the manufacturing of the silica-triptorelin acetate depot, are shown in Figure 1 and Figure 2.

The measured D10, D50 and D90 mean values (±standard deviation) were 1.30 µm ± 0.01 µm, 4.45 µm ± 0.01 µm and 17.12 µm ± 0.30 µm, respectively. The D-values indicate the portion of the particles in the sample are smaller than the given values (e.g., D10 indicates that 10% of the particles are smaller than the given value, and D50-value below 50% and D90-value below 90%, correspondingly). The SEM images show that the silica-triptorelin microparticle bulk consists of smooth spherical microparticles, dimpled spherical microparticles and broken shard-like microparticle shells. The bimodal density distribution measured for the microparticles can be explained by the presence of these different morphologies. The dimpled microparticles and the broken particle shells are smaller in size compared to the smooth spherical microparticles (Figure 2).

The sol-gel parameters as well as the spray-drying conditions are known to impact the final morphology of the microparticles formed by spray-drying [17,18,19]. Therefore, by controlling the sol-gel parameters as well as the spray-drying conditions smooth spherical microparticles can be fabricated from silica nanoparticles by spray-drying [18], which is also shown in the SEM images (Figure 2). It is apparent, that the spray-drying conditions need more optimization work in the future to improve batch consistency by preventing the formation of the broken and dimpled microparticles. These morphologies are most likely formed when the evaporation rate of the solvent, from the droplets that are dried, is too fast and/or the solid content in the droplet is too low [19]. However, regarding the final silica-triptorelin acetate depot formulation, the morphology and particle size does not significantly impact the in vitro and in vivo release kinetics of the final depot. Because the depot formulation, containing the silica-triptorelin microparticles, remains as a single solid entity after administration that gradually surface erodes as the silica matrix dissolves in bodily fluids. This evens out the differences in microparticle morphology. Furthermore, the chemical structure (i.e., degree of condensation) of the silica-triptorelin microparticles has a greater impact on the dissolution rate of the microparticle compared to its size [4,5,6]. Simply by adjusting the condensation degree of the silica matrix, smaller microparticles can be fabricated to dissolve slower compared to larger microparticles and vice versa. 

### 3.2. Viscoelasticity, Dynamic Viscosity and Injectability of the Silica-Triptorelin Acetate Depot

The silica-triptorelin acetate depot was manufactured by mixing spray-dried silica-triptorelin microparticles with a silica sol, which mostly consist of water (ca. 97 wt%) and low concentrations of solid silica nanoparticles (ca. 0.9 wt%). The high-water content silica sol will gradually gel on its own into a hydrogel, but together with the microparticles a stronger physical gel structure is formed. Fundamentally, the silica-triptorelin acetate depot is a particle-reinforced composite material that requires a certain amount of silica nanoparticles in the hydrogel as well as a certain amount of silica microparticles, embedded into the hydrogel, to maintain structural stability of the final depot [20]. The physical gel properties and gelation of sol-gel derived silica gels, used herein to facilitate the administration of the silica-triptorelin microparticles, have been widely reported [21,22,23,24,25]. 

The gel structure of the semisolid silica-triptorelin acetate depot was determined by small angle oscillatory shear measurements presented in Figure 3. The measured G’ values are independent of frequency and are over 10-fold higher than the measured G’’ values. The shear thinning and shear recovery properties of the silica-triptorelin acetate depot are presented in Figure 4. As shear rate is increased the semisolid gel begins to flow and the viscosity of the silica-triptorelin acetate depot rapidly decreases. This is due to the structure of the gel breaking down and aligning along the direction of the shear [26]. After the shear rate (or stress) is decreased and ultimately removed, the structure of the silica-triptorelin acetate depot recovers—seen as an increase in viscosity. Furthermore, manual injection tests were carried out to show that the silica-triptorelin acetate depot could be administered using smaller diameter needle compared to Pamorelin^®^. The silica-triptorelin acetate depot was shown to be injectable through a 25 G needle, whereas Pamorelin^®^ is administered using a 20 G needle. For the pharmacokinetic study, the silica-triptorelin acetate depot was injected using a 23 G needle to ensure low force of injection into the subcutaneous space of the rat.

### 3.3. In Vitro Release of Triptorelin from the Silica-Triptorelin Acetate Depot

The in vitro dissolution results of the silica matrix and resulting triptorelin release are shown in Figure 5. The silica-triptorelin acetate depot contains 12.7 mg of triptorelin acetate per 1 mL injection and has a low initial burst release of triptorelin as the measured 1-h release was below the detection limit of the HPLC method (<0.1 µg/mL). In addition, the release of triptorelin from the silica-triptorelin acetate depot is controlled by the degradation of the silica matrix, which is illustrated in Figure 6. By plotting the cumulative release of triptorelin as a function of cumulative silica degradation, the function is linear and therefore the release rate of triptorelin is constant for 7 days in sink conditions. Tyagi et al. have reported similar properties for an equivalent silica microparticle-silica hydrogel system that was developed for subcutaneous dosing of a therapeutic monoclonal antibody [15].

### 3.4. Pharmacokinetics and Pharmacodynamics of Silica Triptorelin Acetate Depot and Pamorelin^®^

Blood samples of the animal groups (I, II, III, IV) were taken for bioanalysis of triptorelin and testosterone concentration in the circulation. The pharmacokinetic profiles for the administered silica-triptorelin acetate depot and Pamorelin^®^ are shown in Figure 7, whereas the pharmacodynamic profiles are given in Figure 8.

The release of triptorelin from the silica-triptorelin acetate depot showed no initial burst release in neither group of animals (groups II and III). In comparison, the triptorelin release from Pamorelin^®^ showed an initial burst during the first 24 h for both groups I and IV. At the 4-h time-point, the measured triptorelin concentrations for Pamorelin^®^ were ca. 7-fold higher compared to animals that were administered the silica-triptorelin acetate. In addition, the triptorelin plasma concentrations with groups II and III remained stable for the first 2 weeks, and on average higher, compared to groups I and IV, which received Pamorelin^®^. After 2 weeks, the plasma concentrations of triptorelin steadily decreased for each animal group. For 42 days Group III, which received 200 µL injections of the silica-triptorelin acetate depot, showed comparable triptorelin plasma levels as Groups I and IV. Group III also showed triptorelin plasma levels above 0.2 ng/mL (LLOD) up to day 91. In contrast, the lowest triptorelin concentrations were seen for group I, that received 200 µL injections of Pamorelin^®^. A noteworthy observation is the pharmacokinetic similarity between groups II and IV, which received 100 µL injection of the silica-triptorelin acetate depot and the 400 µL injection of Pamorelin^®^, respectively. Arguably, with the silica-triptorelin acetate depot an equivalent drug exposure may be achieved with less triptorelin than Pamorelin^®^ contains (see Table 1). One rationale is that as the initial burst release of triptorelin acetate from the silica-triptorelin acetate depot is lower compared to the corresponding Pamorelin^®^ injection. Therefore, less triptorelin is depleted from the depot within the first hours in vivo. 

The plasma testosterone levels for all groups increase rapidly after administration, reaching concentrations of 9–11 ng/mL (Figure 7). This initial increase in testosterone levels is expected since triptorelin being a LHRH agonist decreases testosterone blood levels through the negative feedback effect [27]. After 24 h the testosterone levels begin to decrease for all groups, reaching therapeutic levels (i.e., human male castrate levels) of 0.2–0.5 ng/mL [28]. For Groups III and IV, which respectively received 200 µL of the silica-triptorelin acetate and 400 µL of Pamorelin^®^, the therapeutic levels of plasma testosterone were maintained up to day 91 (see Table 1). For group I that received 200 µL injections of Pamorelin^®^ the measured testosterone plasma concentrations were the highest out of all the groups. After day 21, the average testosterone levels were near or above 0.5 ng/mL. For Group II, the pharmacokinetic profile for testosterone was equivalent to Group IV. In addition, the suppression of plasma testosterone levels indicates that triptorelin acetate has retained its biological activity in the preparation process, where direct embedment method was used in the preparation of silica-triptorelin acetate depot.

### 3.5. Triptorelin Content in Excised Depot Remnants

After sacrificing the animals on day 91, remnants of silica-triptorelin acetate depots were observed and collected from 11 out of 12 animals (in all animals injected with 1.27 mg triptorelin and five animals injected with 2.54 mg triptorelin) and Pamorelin^®^ remnants were observed and recovered in 3 out 12 animals (in one animal injected with 1.13 mg triptorelin and two animals injected with 2.26 mg triptorelin). The triptorelin content in the Pamorelin^®^ remnants were 7 µg for the lower dose animal and 27 µg and 36 µg for the higher dose animals. In relation to the calculated total content in the injections the amount of triptorelin in the Pamorelin^®^ remnants were 0.6%, 1.2% and 1.7%, respectively. The remaining average amount (n = 6) of triptorelin in the lower dose injections of silica-triptorelin acetate depot, was 85 µg, corresponding to ca. 6.7% of the calculated total content. The average value (n = 5) for the higher dose injection group of silica-triptorelin acetate depot was 263 µg of triptorelin, corresponding to ca. 10.3% of the calculated total content of triptorelin.

The silica-triptorelin acetate is a longer lasting release formulation compared to Pamorelin^®^ as more animals had remnants of the silica-triptorelin acetate depot at the end of the study. The degradation pathways of the test items are different as Pamorelin^®^ is a PLGA-based material that typically undergoes degradation by hydrolysis and cleavage of the polymer backbone into oligomers and then to monomers. The degradation and drug release from PLGA-based formulations are a combination of drug diffusion, surface, and bulk erosion, which involves swelling of the polymer matrix as liquid (e.g., body fluids) penetrates the matrix [29]. In contrast, the degradation and drug release from the silica-triptorelin acetate depot is not as complex compared to PLGA. The silica matrix of the silica depot undergoes surface erosion and the degradation and drug release from the silica matrix is controlled by the dissolution of silica into silicic acid. In vivo silicic acid is excreted out with urine and the dissolution of silica is controlled by the local silica concentration at the site of injection. 

The silica-triptorelin acetate depot remnants were observed as single solid entities, which could be removed accurately (Figure 9). The Pamorelin^®^ remnants were more difficult to remove as the injected microparticles were spread-out in the tissue. Therefore, the remaining Pamorelin^®^ was difficult to collect completely, which can also explain why less triptorelin was measured for the Pamorelin^®^ remnants compared to the silica-triptorelin acetate depot remnants. This structural difference of these two test items can also explain, why the silica-triptorelin acetate depot remnants were observed to degrade slower compared to Pamorelin^®^—a single solid entity has less reactive surface compared to separated microparticles.

## 4. Conclusions

The developed silica-triptorelin acetate depot showed promising pharmacodynamic and pharmacokinetic results in the rat and further development of the silica-triptorelin acetate depot could yield a promising alternative for Pamorelin^®^ in the treatment of hormone-dependent prostate cancer. 

It was shown by pharmacodynamic effect (suppression of plasma testosterone levels) that triptorelin acetate retained its biological activity after manufacturing the silica-triptorelin acetate microparticles and embedding said microparticles into a silica hydrogel. The resulting silica-triptorelin acetate depot was injectable with 23 G and 25 G needles, which is beneficial in terms of injectability, and patient compliance compared to Pamorelin^®^ that is administered using a 20 G needle. The remnant analysis showed that the hydrogel-based dosage form of silica-triptorelin acetate depot retained its 3 D structure in tissue and it was not spread-out as the microparticles of Pamorelin^®^.

In vivo pharmacokinetics showed that on average injections of silica-triptorelin acetate depot formulation gave 5-fold lower Cmax values than the corresponding Pamorelin^®^ injections. The silica-triptorelin acetate depot also showed long lasting triptorelin release comparable to the commercially available Pamorelin^®^.

## Figures and Tables

**Figure 1 nanomaterials-11-01578-f001:**
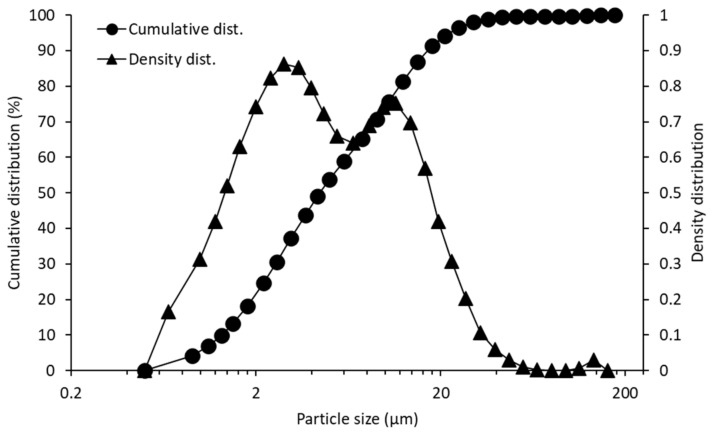
Cumulative and density distribution of silica-triptorelin acetate microparticles. Data represents three replicate measurements of one lot of silica-triptorelin acetate microparticles.

**Figure 2 nanomaterials-11-01578-f002:**
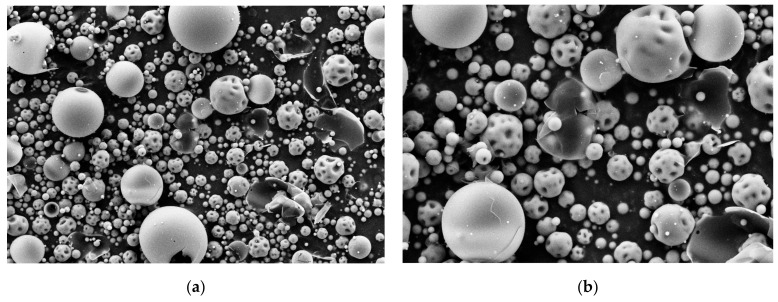
SEM images of silica-triptorelin acetate microparticles: (**a**) Magnification 2500×; (**b**) Magnification 5000×.

**Figure 3 nanomaterials-11-01578-f003:**
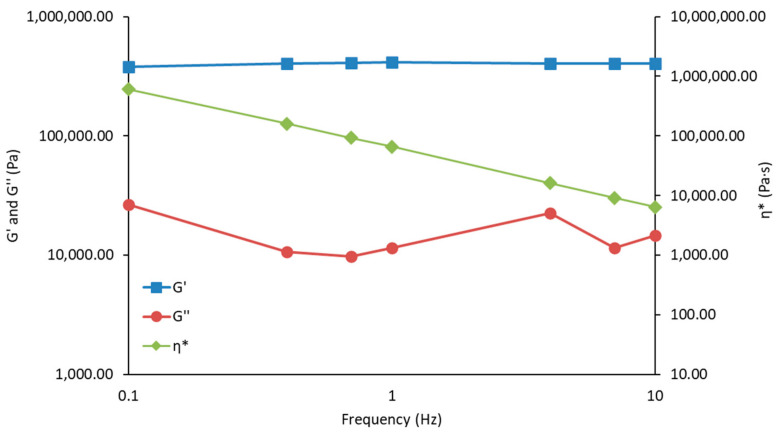
Storage modulus (G’), loss modulus (G″) and complex viscosity (η*) of the silica-triptorelin acetate depot confirming the gel structure of the semi-solid depot.

**Figure 4 nanomaterials-11-01578-f004:**
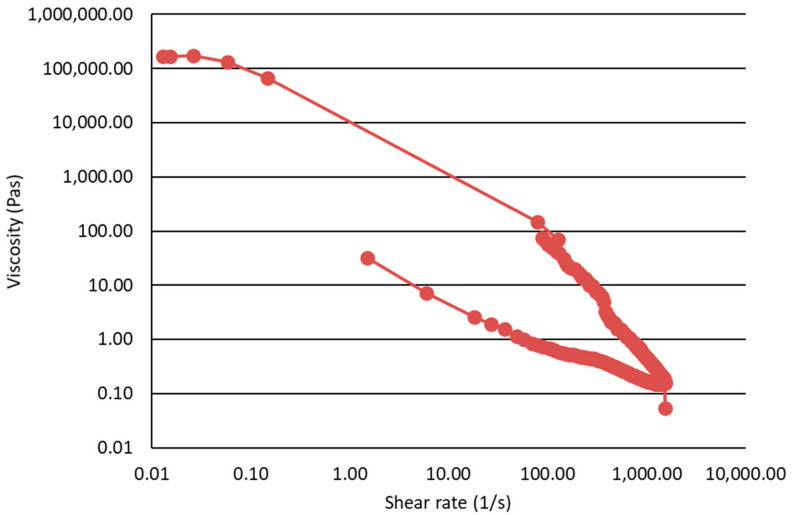
Dynamic viscosity of the silica-triptorelin acetate depot illustrating the shear recovery behavior of the depot.

**Figure 5 nanomaterials-11-01578-f005:**
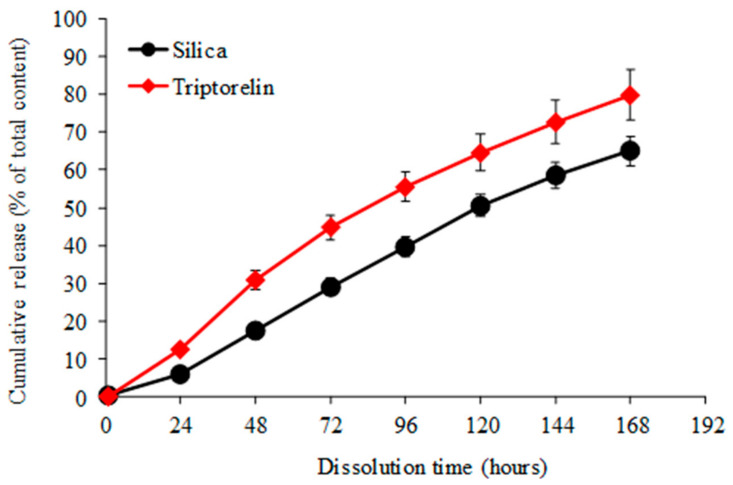
Cumulative in vitro silica degradation and resulting release of triptorelin (in sink conditions) in 50 mM TRIS buffer, containing 0.01 % (*v*/*v*) TWEEN^®^ 80, (pH 7.4 at 37 °C). Data represents mean values of three replicate dissolutions and standard deviation.

**Figure 6 nanomaterials-11-01578-f006:**
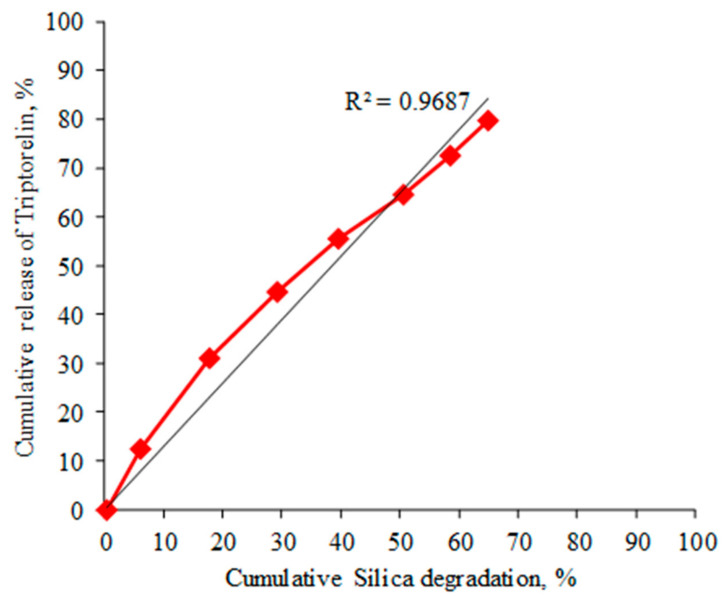
Correlation between cumulative triptorelin release and cumulative degradation of the silica matrix for the silica-triptorelin acetate depot.

**Figure 7 nanomaterials-11-01578-f007:**
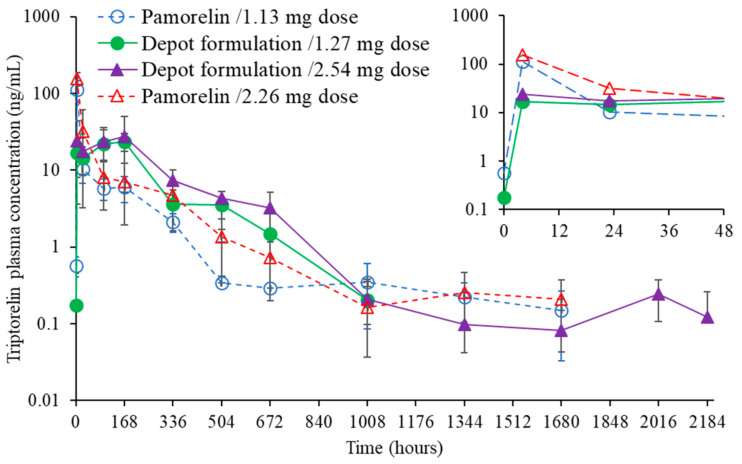
Plasma concentration of triptorelin before and after the injection of Pamorelin^®^ or silica-triptorelin acetate depot under two-dose regimens. Data represents mean values with standard deviation as error bars. Overlaid insert graph shows zoom in on measured mean triptorelin plasma concentrations up to 48 h (top right corner).

**Figure 8 nanomaterials-11-01578-f008:**
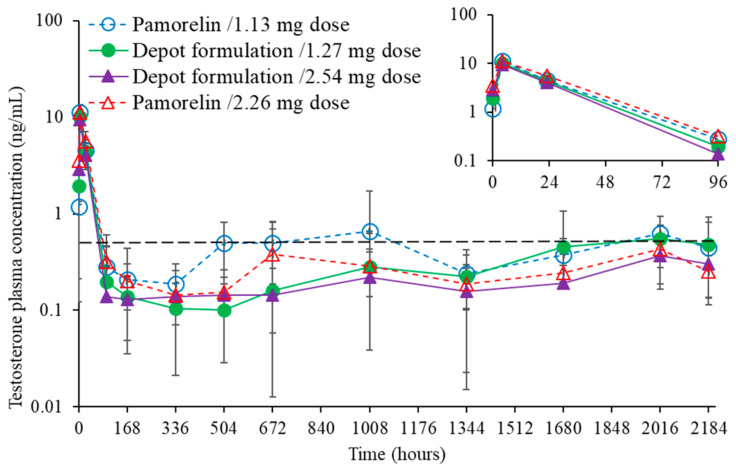
Plasma concentration of testosterone before and after the injection of Pamorelin^®^ or silica-triptorelin acetate depot under two-dose regimens. Castration level for human males (0.5 ng/mL) is marked with long-dash line. Data represents mean values with standard deviation. Overlaid insert graph shows measured mean testosterone plasma concentrations up to 96 h (top right corner).

**Figure 9 nanomaterials-11-01578-f009:**
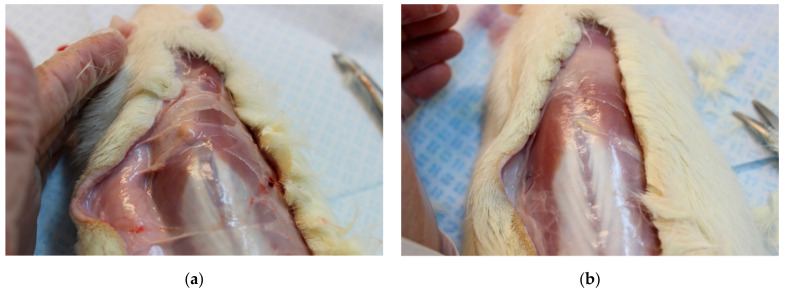
Photographs of excised remnants at day 91 after necropsy: (**a**) silica-triptorelin acetate depot remnant from a group III animal (**b**) Pamorelin^®^ remnant from a group IV animal.

**Table 1 nanomaterials-11-01578-t001:** Comparison of pharmacokinetic parameters for triptorelin and resulting impact on testosterone.

Analyte	Parameter	Pamorelin^®^		Silica-Triptorelin Depot	
		1.13 mg Dose ^1^	2.26 mg Dose ^1^	1.27 mg Dose ^1^	2.54 mg Dose ^1^
Triptorelin	C_max_ (ng/mL)	116 ± 34	159 ± 31	25.2 ± 10.2	34.2 ± 22.5
	AUC_last_ (h*ng/mL)	3760 ± 961	7002 ± 2169	7366 ± 958	10 380 ± 2782
	CL (L/day)	7.2	7.7	4.1	5.9
Testosterone	C_max_ (ng/mL)	11.4 ± 2.3	11.4 ± 2.0	10.3 ± 2.0	9.7 ± 2.3
	AUC_last_ (h*ng/mL)	1471 ± 612	1000 ± 209	1084 ± 632	721 ± 370
	C_last_ (ng/mL)	0.58 ± 0.48	0.27 ± 0.12	0.59 ± 0.34	0.33 ± 0.19
	T_last_ (day)	91	91	91	91

^1^ Dose of triptorelin that was administered. Data represents average values ± standard deviation.

## Data Availability

Not applicable.

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
