# Peer review of "Sustained In-Vivo Release of Triptorelin Acetate from a Biodegradable Silica Depot: Comparison to Pamorelin® LA"

_nanomaterials, 2021, doi:10.3390/nano11061578_

Round 1
Reviewer 1 Report
Dear Authors,
I have carefully read the manuscript on Sustained In-Vivo Release of Triptorelin Acetate from a Biodegradable Silica Depot: Comparison to Pamorelin® LA.
The paper contains numerous cognitive elements and interesting preparation, but in my opinion is not suitable for the journal Nanomaterials. You are working on inorganic silica particles that do not have the size or behavior of characteristic for nanomaterials. The introduced active substance - tryptorelin (in the form of tryptorelin acetate), which is a synthetic analogue of gonadotropin-releasing hormone (GnRH) does not have nanometric size but molecular size.
I consider the process of obtaining and studying the release kinetics of the active substance itself as correct, but there is no strong justification why you count your material, which in DLS shows micrometric size, as nanometric size? Silica obtained in the developed way does not even have mesopores (2-50nm), which could be classified as nanomaterial features.
In my opinion, the work should be sent to a journal dealing with pharmacy possibly biomaterials.
Author Response
Dear Reviewer, please find our answers below:
The resulting composite material (“Silica depot”) is not fully a nanomaterial (and naturally not a nanoparticle), but it comprises of silica nanoparticles (in addition to silica microparticles and water), and the role of the nanoparticles is crucial to achieve the desired properties. If only silica microparticles are suspended in water, they do not form a stable suspension (because they are not colloidal), i.e., they will sediment in few hours to days, and if another, more viscous liquid is used, it would take longer, but still the microparticles would sediment.
The key thing is that the resulting composite material is stable and injectable, and it was achieved by adding a small number of silica nanoparticles (in the form of a silica sol) into a silica microparticle suspension, and by that addition a non-stable silica microparticle suspension turned into a stable and non-flowing gel (non-flowing gel structure prevents sedimentation of silica microparticles). This “Silica depot” is stable and non-flowing at rest (e.g., in a pre-filled syringe), but shear-thinning under shear (e.g., during injection through a thin needle, practical examples in this paper were 23G and 25G needles), and it turns again into a non-flowing gel after the injection and takes a 3D form. These things are described briefly in Abstract, and in more detail in Materials & Methods (page 3, last paragraph) and in Chapter 3.2.
In addition, the silica microparticles are actually clusters of nanoparticle aggregates, which have been prepared from silica sols, in other words from silica nanoparticles and silica nanoparticle aggregates in a liquid, and the aggregation of nanoparticles is controlled (with help of formulation details, e.g., water/alkoxide ratios, pH, 2-step water addition etc., and with processing details, e.g., time window for spray-drying (formulation details for the peptide are described in Materials and Methods) in a manner, which results in silica microparticles that have such a small pore size, that only the dissolution of silica in body fluids controls the release of any API, in this case a peptide, i.e., the role of the pores in the release is minimized. The main property of the silica microparticles, controlled release of the peptide, is tested with a direct and quantitative release experiments (in vitro in simulated physiological liquids in sink conditions) because gas adsorption/desorption is not accurate enough, and nor are, e.g., electron micrographs, and neither of them is quantitative enough (and certainly not direct methods) to see the role of the silica dissolution vs. role of the pores. The mechanism of release is seen by measuring both silica dissolution rate and peptide release rate simultaneously, i.e., it is seen how the dissolution rate of silica microparticles control the peptide release, and whether the pores have any role in it. These things and silica dissolution-controlled release of the peptide are discussed at general level in Introduction (e.g., comparison between mesoporous silica and encapsulation/embedment method used in this paper), and practical results are in Chapter 3.3, both in the text and in Figures 5 and 6.
Reviewer 2 Report
Sustained In-Vivo Release of Triptorelin Acetate from a Biode-2 gradable Silica Depot: Comparison to Pamorelin® LA by Forsback etal demonstrated the Triptorelin acetate in in vitro and in vivo release in rats.
Major comment: It an interesting study, however it is important to evaluate the toxicity in blood serum and also in in vitro models.
Author Response
The safety and tolerability of biodegradable silica has been extensively researched in the past. The cytotoxic potential of SiO2-R52.5 was assessed in the mouse connective tissue cell line L929 in vitro. SiO2-R52.5 was extracted with RPMI 1640 cell culture medium supplemented with 10% (v/v) foetal calf serum. The cells were incubated with SiO2-R52.5 extract concentrations 3%, 10%, 30%, and 100% (v/v), the corresponding concentrations of a positive extractant control (reference standard according to BS5736-10), with the extractant medium control alone, and with a negative extract control (high-density polyethylene) for 24 h. The cells were labelled with the XTT {sodium 3′-[1-[(phenylamino)-carbonyl]-3,4-tetrazolium]-bis(4-methoxy-6-nitro) benzene sulfonic acid hydrate} reagent to form an orange water soluble dye by active mitochondria. The positive control showed a distinct dose-dependent reduction in cell viability and proliferation. No cytotoxic effects were observed with any of the SiO2-R52.5 extract concentrations. No relevant difference between the extractant control and the negative control could be observed either. In conclusion, the SiO2-R52.5 extract did not possess any cytotoxic potential in L929 cells. (Heppenheimer A. Cytotoxicity assay in vitro: Evaluation of materials for medical devices (XTT test) with SiO2-R52.5. REPORT, RCC–CCR Study Number 1077500. RCC, Itingen, Switzerland, 2007.)
Reviewer 3 Report
The aim of the study was to adapt the silica-based composite, comprising of silica microparticles embedded in a silica hydrogel, for controlled and sustained delivery of triptorelin acetate (a decapeptide). The work is clear and the method is well explained. In table 1 the standard deviation for the values should be added. Minor revisions of English should be made. Overall, if these questions are addressed they can be published.
Author Response
Dear reviewer,
W have added the standard deviations to Table 1.
Reviewer 4 Report
The manuscript entitled "Sustained In-Vivo Release of Triptorelin Acetate from a Biode-2 gradable Silica Depot: Comparison to Pamorelin® LA" by Ari-Pelka et. al. has been revised. In this paper was studied the synthesis and characterization of silica-Triptorelin microparticles for rats subcutaneous administration. The pharmacokinetics studies entail promising resultsgiving 5-fold lower Cmax values that the corresponding Parmolein injections and also sustained triptorelin release. All these results are interesting. The paper is well-written and the results obtained are well-supported by the experimental data and the manuscript well meets the scope requirement of the journal. However, some questions should be answered before paper publication in Nanomaterials:
-The introduction of the paper is complete and exhaustive. However, the reason by which Triptorelin is employed instead of either API is missing.
-Description of some techniques and protocols in section Materials and Methods is too long. It should be shortened and complete in the Supplementary information of the paper.
-The stability of the synthesized silica-Triptorelin microparticles with time must be provided. For instance, UV-vis spectrophotometry can be used for measuring stability.
-The authors state:" Furthermore, the chemical structure (i.e. , degree of condensation) of the silica-triptorelin microparticles has a great impact on the disolution rate of the microparticle compared to its size". Have the authors determine the degree of condensation of the silica-Triptorelin in several preparations to confirm this assumption?
-lines 295-296, page 7. Please specify the unit of measurement for the size of D1, D50 and D90. Moreover, D1, D50 and D90 must be defined.
-Line 150, page 3. "Just by...incorporating" delete the space.
Author Response
Dear reviewer,
we added our answers below the questions.
-The introduction of the paper is complete and exhaustive. However, the reason by which Triptorelin is employed instead of either API is missing.
Answer: We wanted a model molecule for controlled delivery of peptides. Triptorelin acetate was chosen to show the potential of the silica technology for controlled delivery of peptides due to good pharmacodynamic and pharmacokinetic modelling This rationale was added to the introduction.
-Description of some techniques and protocols in section Materials and Methods is too long. It should be shortened and complete in the Supplementary information of the paper.
Answer: We feel this is up to the editor to decide and it can be done so, if decided.
-The stability of the synthesized silica-Triptorelin microparticles with time must be provided. For instance, UV-vis spectrophotometry can be used for measuring stability.
Answer: The stability of triptorelin is shown directly biologically in the pharmacodynamics results in this paper. The chemical analysis are indirect methods that could provide further support, but they were not evaluated in this study. The in vivo dose was calculated based on the API payload in the microparticles and the expected pharmacokinetic and pharmacodynamic profiles were achieved.
-The authors state:" Furthermore, the chemical structure (i.e. , degree of condensation) of the silica-triptorelin microparticles has a great impact on the dissolution rate of the microparticle compared to its size". Have the authors determined the degree of condensation of the silica-Triptorelin in several preparations to confirm this assumption?
Answer: In this study the degree of condensation was not evaluated for the microparticles. However, it can be quantified by NMR measurements, and it has been shown earlier that the degree of condensation impacts/determines the dissolution rate [Reeta Viitala, Mika Jokinen, Sirkka Liisa Maunu, Harry Jalonen, Jarl B. Rosenholm, Chemical characterization of bioresorbable sol–gel derived SiO2 matrices prepared at protein-compatible pH. Journal of Non-Crystalline Solids,Volume 351, Issues 40–42,2005,pages 3225-3234,https://doi.org/10.1016/j.jnoncrysol.2005.08.023].
-lines 295-296, page 7. Please specify the unit of measurement for the size of D1, D50 and D90. Moreover, D1, D50 and D90 must be defined.
Answer: The units were added to the manuscript and the D-values are defined in the manuscript after the results.
-Line 150, page 3. "Just by...incorporating" delete the space.
Answer: The space was deleted.
Reviewer 5 Report
Noppari et al demonstrate an in vivo pharmacokinetics and 16 pharmacodynamics of the silica-triptorelin acetate depot and Pamorelin®in Sprague-Dawley male rats. Experimental results are interesting and well characterized. I recommend it publication after minor revision as follows.
- The biotoxicity of the silica-triptorelin ac- 14etate microparticles should be investigated.
- It is better to study how these particles remove from rats body.
Author Response
Dear reviewer,
we have added our answers below the questions.
1.The biotoxicity of the silica-triptorelin ac- 14etate microparticles should be investigated.
Answer: In the preclinical study in the rat, the tolerability and safety was evaluated. No clinical adverse signs were seen during the study (local or systemic). In general, the tolerability and safety of biodegradable silica has been widely investigated in the past in GLP-tox studies in the rat and beagle-dog.
2. It is better to study how these particles remove from rats body.
Answer: The silica particles dissolve in bodily fluids. The dissolution product is silicic acid which is excreted out through kidneys in urine.
Round 2
Reviewer 1 Report
Dear Authors, Thank you for your explanations. Unfortunately I still don't see a clear exposition of the role of silica nanoparticles in the experiment. The authors write that silica agglomerates (they even complement the particle size, proving that in the experiment the key role is played by particles that are micrometric) and as nanoparticles cannot exist in the hinge, so wouldn't it make more sense to send the paper to a biomaterials/pharmaceutical journal. The scope of the journal includes according to the publisher: Nanomaterials are materials with typical size features in the lower nanometer size range and characteristic mesoscopic properties; for example quantum size effects. These properties make them attractive objects of fundamental research and potential new applications. The scope of Nanomaterials covers the preparation, characterization and application of all nanomaterials. The following examples may provide a guide to what will be covered (not exclusive): o Nanomaterials: nanoparticles, coatings and thin films, inorganic-organic hybrids and composites (i.e. MOFs), membranes, nano-alloys, quantum dots, self-assemblies, graphene, nanotubes, etc o Methodologies: Synthesis of organic, inorganic, and hybrid nanomaterials, characterization of mesoscopic properties, modelling of nanomaterials and/or mesoscopic effects o Applications: any application of new nanomaterials or new application of nanomaterials If in the paper (not in the explanations) there are arguments confirming the role of nanosilica and the exception of its behavior. In my opinion, the manuscript will have a justification to be sent and directed to publication in NanomaterialsAuthor Response
Dear reviewer.
Nanomaterials have been used in the composite. Both microparticles and nanoparticles play a key role. Microparticles play a key role in the controlled drug release, whereas the nanoparticles play a key role in the rheological properties (stable non-flowing gel at rest, flow under shear i.e. shear thinning, recovery of gel structure after injection). These justifications have been described in the paper. We feel we cannot explain any further. We feel that the editors must now decide whether the work is in scope.
Reviewer 2 Report
Now the manuscript reads good
Author Response
Dear reviewer.
Excellent!
Reviewer 4 Report
The paper has been revised again for possible publication in Nanomaterials. The authors have performed some of the revision required. However, there are some points that need clarification:
-Description of some techniques and protocols in section Materials and Methods is too long. It should be shortened and complete in the Supplementary information of the paper.
-A reference must be added to justify the authors state: " Furthermore, the chemical structure (i.e. , degree of condensation) of the silica-triptorelin microparticles has a great impact on the dissolution rate of the microparticle compared to its size".
Author Response
Dear reviewer,
-We are happy to shorten the descriptions, when we receive instructions from the editor on how to do it. What should be taken out.
-Thank you for the observation. The existing references of our earlier work with corresponding silica matrices [4-6] regarding the chemical structure, degree of condensation and dissolution rate have been added to the statement.
This manuscript is a resubmission of an earlier submission. The following is a list of the peer review reports and author responses from that submission.